# Quantum Optical Experiments Modeled by Long Short-Term Memory

## Abstract

We demonstrate how machine learning is able to model experiments in quantum physics. Quantum entanglement is a cornerstone for upcoming quantum technologies such as quantum computation and quantum cryptography. Of particular interest are complex quantum states with more than two particles and a large number of entangled quantum levels. Given such a multiparticle high-dimensional quantum state, it is usually impossible to reconstruct an experimental setup that produces it. To search for interesting experiments, one thus has to randomly create millions of setups on a computer and calculate the respective output states. In this work, we show that machine learning models can provide significant improvement over random search. We demonstrate that a long short-term memory (LSTM) neural network can successfully learn to model quantum experiments by correctly predicting output state characteristics for given setups without the necessity of computing the states themselves. This approach not only allows for faster search but is also an essential step towards automated design of multiparticle high-dimensional quantum experiments using generative machine learning models.

## 1 Introduction

In the past decade, artificial neural networks have been applied to a plethora of scientific disciplines, commercial applications, and every-day tasks with outstanding performance in, e.g., medical diagnosis, self-driving, and board games (Esteva et al., 2017; Silver et al., 2017). In contrast to standard feedforward neural networks, long short-term memory (LSTM) (Hochreiter, 1991; Hochreiter & Schmidhuber, 1997) architectures have recurrent connections, which allow them to process sequential data such as text and speech (Sutskever et al., 2014).

Such sequence-processing capabilities can be particularly useful for designing complex quantum experiments, since the final state of quantum particles depends on the sequence of elements, i.e. the experimental setup, these particles pass through. For instance, in quantum optical experiments, photons may traverse a sequence of wave plates, beam splitters, and holographic plates. High-dimensional quantum states are important for multiparticle and multisetting violations of local realist models as well as for applications in emerging quantum technologies such as quantum communication and error correction in quantum computers (Shor, 2000; Kaszlikowski et al., 2000).

Already for three photons and only a few quantum levels, it becomes in general infeasible for humans to determine the required setup for a desired final quantum state, which makes automated design procedures for this inverse problem necessary. One example of such an automated procedure is the algorithm MELVIN (Krenn et al., 2016), which uses a toolbox of optical elements, randomly generates sequences of these elements, calculates the resulting quantum state, and then checks whether the state is interesting, i.e. maximally entangled and involving many quantum levels. The setups proposed by MELVIN have been realized in laboratory experiments (Malik et al., 2016; Erhard et al., 2018b). Recently, also a reinforcement learning approach has been applied to design new experiments (Melnikov et al., 2018).

Inspired by these advances, we investigate how LSTM networks can learn quantum optical setups and predict the characteristics of the resulting quantum states. We train the neural networks using millions of setups generated by MELVIN. The huge amount of data makes deep learning approaches the first choice. We use cluster cross validation (Mayr et al., 2016) to evaluate the models.

## 2 METHODS

### 2.1 TARGET VALUES

Let us consider a quantum optical experiment using three photons with orbital angular momentum (OAM) (Yao & Padgett, 2011; Erhard et al., 2018a). The OAM of a photon is characterized by an integer whose size and sign represent the shape and handedness of the photon wavefront, respectively. For instance, after a series of optical elements, a three particle quantum state may have the following form:

$$|\Psi\rangle = \frac{1}{2} \left( |0, 0, 0\rangle + |1, 0, 1\rangle + |2, 1, 0\rangle + |3, 1, 1\rangle \right). \tag{1}$$

This state represents a physical situation, in which there is 1/4 chance (modulus square of the amplitude value 1/2) that all three photons have OAM value 0 (first term), and a 1/4 chance that photons 1 and 3 have OAM value 1, while photon 2 has OAM value 0 (second term), and so on for the two remaining terms.

We are generally interested in two main characteristics of the quantum states: (1) Are they maximally entangled? (2) Are they high-dimensional? The dimensionality of a state is represented by its Schmidt rank vector (SRV) (Huber & de Vicente, 2013; Huber et al., 2013). State $|\Psi\rangle$ is indeed maximally entangled because all terms on the right hand side have the same amplitude value. Its SRV is (4,2,2), as the first photon is four-dimensionally entangled with the other two photons, whereas photons two and three are both only two-dimensionally entangled with the rest.

A setup is labeled "positive" ($y_E = 1$) if its output state is maximally entangled and if the setup obeys some further restrictions, e.g., behaves well under multi-pair emission, and otherwise labeled "negative" ($y_E = 0$). The target label capturing the state dimensionality is the SRV $y_{SRV} = (n, m, k)^\top$. We train LSTM networks to directly predict these state characteristics (entanglement and SRV) from a given experimental setup without actually predicting the quantum state itself.

### 2.2 LOSS FUNCTION

For classification, we use binary cross entropy (BCE) in combination with logistic sigmoid output activation for learning. For regression, it is always possible to reorder the photon labels such that the SRV has entries in non-increasing order. An SRV label is thus represented by 3-tuple $y_{SRV} = (n, m, k)^\top$ which satisfies $n \geq m \geq k$. With slight abuse of notation, we model $n \sim \mathcal{P}(\lambda)$ as a Poisson-distributed random variable and $m \sim \mathcal{B}(n, p), k \sim \mathcal{B}(m, q)$ as Binomials with ranges $m \in \{1, \ldots n\}$ and $k \in \{1, \ldots, m\}$ and success probabilities $p$ and $q$, respectively. The resulting log-likelihood objective (omitting all terms not depending on $\lambda, p, q$) for a data point $x$ with label $(n, m, k)^\top$ is

$$\ell(\hat{\lambda}, \hat{p}, \hat{q} \mid x) = n \log \hat{\lambda} - \hat{\lambda} + m \log \hat{p} + (n - m) \log(1 - \hat{p}) + k \log \hat{q} + (m - k) \log(1 - \hat{q}) \tag{2}$$

where $\hat{\lambda}, \hat{p}, \hat{q}$ are the network predictions (i.e. functions of $x$) for the distribution parameters of $n, m, k$ respectively. The Schmidt rank value predictions are $\hat{n} = \hat{\lambda}, \hat{m} = \hat{p}\hat{\lambda}, \hat{k} = \hat{p}\hat{q}\hat{\lambda}$. To see this, we need to consider the marginals of the joint probability mass function

$$f(n, m, k) = \frac{\lambda^n e^{-\lambda}}{n!} \binom{n}{m} p^m (1 - p)^{n-m} \binom{m}{k} q^k (1 - q)^{m-k}. \tag{3}$$

To obtain the marginal distribution of $m$, we can first sum over all possible $k$, which is easy. To sum out $n$ we first observe that $\binom{n}{m} = 0$ for $n < m$, i.e. the first $m$ terms are zero and we may write

$$f(m) = \sum_{n=0}^{\infty} f(n, m) = \sum_{n=0}^{\infty} f(m + n, m) \tag{4}$$

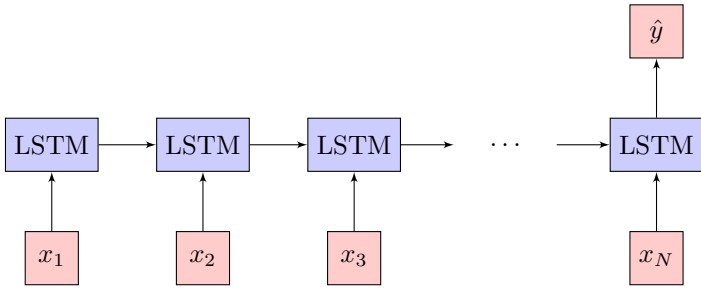

Figure 1: Sequence processing model for a many-to-one mapping. The target value $\hat{y}$ can be either an estimate for $y_{\mathrm{E}}$ (entanglement classification) or $y_{\mathrm{SRV}}$ (SRV regression).

capturing only non-zero terms. It follows that

$$
\begin{aligned}
f(m) &= \sum_{n=0}^{\infty} \frac{\lambda^{n+m} e^{-\lambda}}{(n+m)!} \binom{n+m}{m} p^m (1-p)^n \\
&= e^{-\lambda} p^m \lambda^m \sum_{n=0}^{\infty} \frac{\lambda^n (1-p)^n}{(n+m)!} \binom{n+m}{m} \\
&= \frac{e^{-\lambda} p^m \lambda^m}{m!} \sum_{n=0}^{\infty} \frac{\lambda^n (1-p)^n}{n!} = \frac{e^{-p\lambda} (p\lambda)^m}{m!},
\end{aligned}
\tag{5}
$$

which is $\mathcal{P}(p\lambda)$-distributed. Using the same argument for $k$ we get that the marginal of $k$ is $\mathcal{P}(pq\lambda)$-distributed. The estimates $\hat{n}, \hat{m}, \hat{k}$ are obtained by taking the means of their respective marginals.

### 2.3 NETWORK ARCHITECTURE

The used sequence processing model is depicted in Figure 1. We train two networks, one for entanglement classification (target $y_{\mathrm{E}}$), and one for SRV regression (target $y_{\mathrm{SRV}}$). The reason why we avoid multitask learning in this context is that we do not want to incorporate correlations between entanglement and SRV into our models. For instance, the SRV (6,6,6) was only observed in non-maximally entangled samples so far, which is a perfect correlation. This would cause a multitask network to automatically label such a sample as negative only because of its SRV. By training separate networks we lower the risk of incorporating such correlations.

A setup of $N$ elements is being fed into a network by its sequence of individual optical components $x = (x_1, x_2, ..., x_N)^{\top}$, where in our data $N$ ranges from 6 to 15. We use an LSTM with 2048 hidden units and a component embedding space with 64 dimensions. The component embedding technique is similar to word embeddings (Mikolov et al., 2013).

## 3 EXPERIMENTS

### 3.1 DATASET

The dataset produced by MELVIN consists of 7,853,853 different setups of which 1,638,233 samples are labeled positive. Each setup consists of a sequence $x$ of optical elements, and the two target values $y_{\mathrm{E}}$ and $y_{\mathrm{SRV}}$. We are interested in whether the trained model is able to extrapolate to unseen SRVs. Therefore, we cluster the data by leading Schmidt rank $n$. Figure 2 shows the the number of positive and negative samples in the data set for each $n$.

### 3.2 WORKFLOW

All samples with $n \geq 9$ are moved to a special *extrapolation set* consisting of only 1,754 setups (gray cell in Table 1). The remainder of the data, i.e. all samples with $n < 9$, is then divided into a training set and a conventional *test set* with 20 % of the data drawn at random (iid). This workflow is shown in Figure 3.

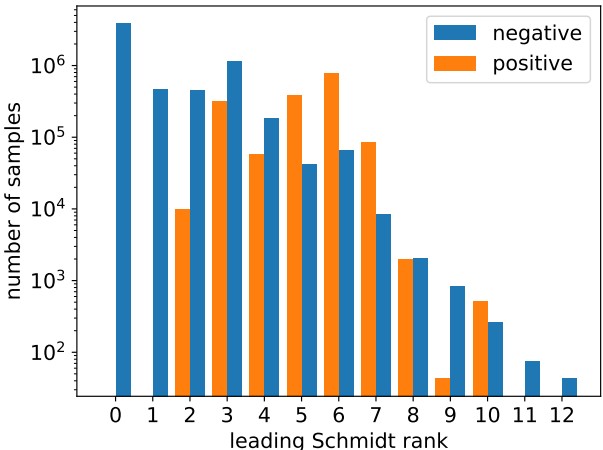

Figure 2: Negative and positive samples in the data set as a function of the leading Schmidt rank $n$.

| 0,1 / 0,1 | 2 | 3 | 4 | 5 | 6 | 7 | 8 | 9-12 |
|---|---|---|---|---|---|---|---|---|

Table 1: Cluster cross validation folds (0-8) and extrapolation set (9-12) characterized by leading Schmidt rank $n$. Samples with $n = 0$ and samples with $n = 1$ are combined and then split into two folds (0,1) at random.

The test set is used to estimate the conventional generalization error, while the extrapolation set is used to shed light on the ability of the learned model to perform on higher Schmidt rank numbers. If the model extrapolates successfully, we can hope to find experimental setups that lead to new interesting quantum states.

Cluster cross validation (CCV) is an evaluation method similar to standard cross validation. Instead of grouping the folds iid, CCV groups them according to a clustering method. Thus, CCV removes similarities between training and validation set and simulates situations in which the withheld folds have not been obtained yet, thereby allowing us to investigate the ability of the network to discover these withheld setups. We use CCV with nine folds (white cells in Table 1). Seven of these folds correspond to the leading Schmidt ranks $2, \ldots, 8$. The samples with $n = 1$ (not entangled) and $n = 0$ (not even a valid three-photon state) are negative by definition. These samples represent special cases of $y_{\mathrm{E}} = 0$ setups and it is not necessary to generalize to these cases without training on them. Therefore, the 4,300,268 samples with $n < 2$ are divided into two folds at random such that the model will always see some of these special samples while training.

### 3.3 RESULTS

Let us examine if the LSTM network has learned something about quantum physics. A good model will identify positive setups correctly while discarding as many negative setups as possible. This behavior is reflected in the metrics *true positive rate* $\mathrm{TPR} = \mathrm{TP}/(\mathrm{TP} + \mathrm{FN})$ and *true negative rate* $\mathrm{TNR} = \mathrm{TN}/(\mathrm{TN} + \mathrm{FP})$, with TP, TN, FP, FN the true positives, true negatives, false positives, false negatives, respectively. A metric that quantifies the success rate within the positive predictions is the *hit rate* (i.e. precision or positive predicted value), defined as $\mathrm{HR} = \mathrm{TP}/(\mathrm{TP} + \mathrm{FP})$ (Simm et al., 2018).

For each withheld CCV fold $n$, we characterize a setup to be "interesting" when it fulfills the following two criteria: (i) It is classified positive ($\hat{y}_{\mathrm{E}} > \tau$) with $\tau$ the classification threshold of the sigmoid output activation. (ii) The SRV prediction $\hat{y}_{\mathrm{SRV}} = (\hat{n}, \hat{m}, \hat{k})^{\top}$ is such that there exists a $y_{\mathrm{SRV}} = (n, m, k)^{\top}$ with $\|y_{\mathrm{SRV}} - \hat{y}_{\mathrm{SRV}}\|_2 < r$. We call $r$ the SRV radius. We denote samples which are classified as interesting (uninteresting) and indeed positive (negative) as true positives

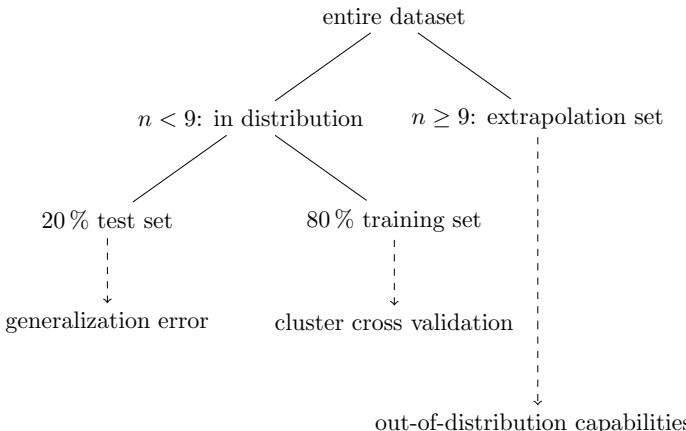

Figure 3: Workflow. We split the entire data by their leading Schmidt rank $n$. All samples with $n \geq 9$ constitute the extrapolation set, which we use to explore the out-of-distribution capabilities of our model. For the remaining samples (i.e. $n < 9$) we make a random test split at a ratio of $1/4$. The test set is used to estimate the conventional generalization error of our model. We use the training set to perform cluster cross validation.

(negatives). And we denote samples which are classified as interesting (uninteresting) and indeed negative (positive) as false positives (false negatives).

We employ stochastic gradient descent for training the LSTM network with momentum 0.5 and batch size 128. We sample mini-batches in such a way that positive and negative samples appear equally often in training. For balanced SRV regression, the leading Schmidt rank vector number $n$ is misused as class label. The models were trained using early stopping after 40000 weight update steps for the entanglement classification network and 14000 update steps for the SRV regression network. Hyperparameter search was performed in advance on a data set similar to the training set.

Figure 4 shows the TNR, TPR, and rediscovery ratio for sigmoid threshold $\tau = 0.5$ and SRV radius $r = 3$. The rediscovery ratio is defined as the number of distinct SRVs, for which at least 20% of the samples are rediscovered by our method, i.e. identified as interesting, divided by the number of distinct SRVs in the respective cluster. The TNR for fold 0,1 is 0.9996, and the hit rate HR on the extrapolation set 9-12 is 0.659. Error bars in Figure 4 and later in the text are 95 % binomial proportion confidence intervals. Model performance depends heavily on parameters $\tau$ and $r$. Figure 5 shows the "beyond distribution" results for a variety of sigmoid thresholds and SRV radii.

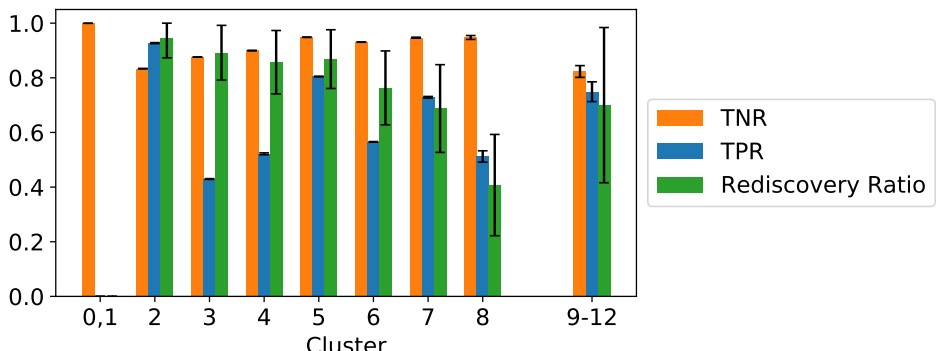

Figure 4: True negative rate (TNR), true positive rate (TPR), rediscovery ratio of the LSTM network using cluster cross validation for different folds 0-8. True negative rates are high for all validation folds. All metrics are good for the extrapolation set 9-12, demonstrating that the models perform well on data beyond the training set distribution, covering only Schmidt rank numbers 0-8. Error bars represent 95 % binomial proportion confidence intervals.

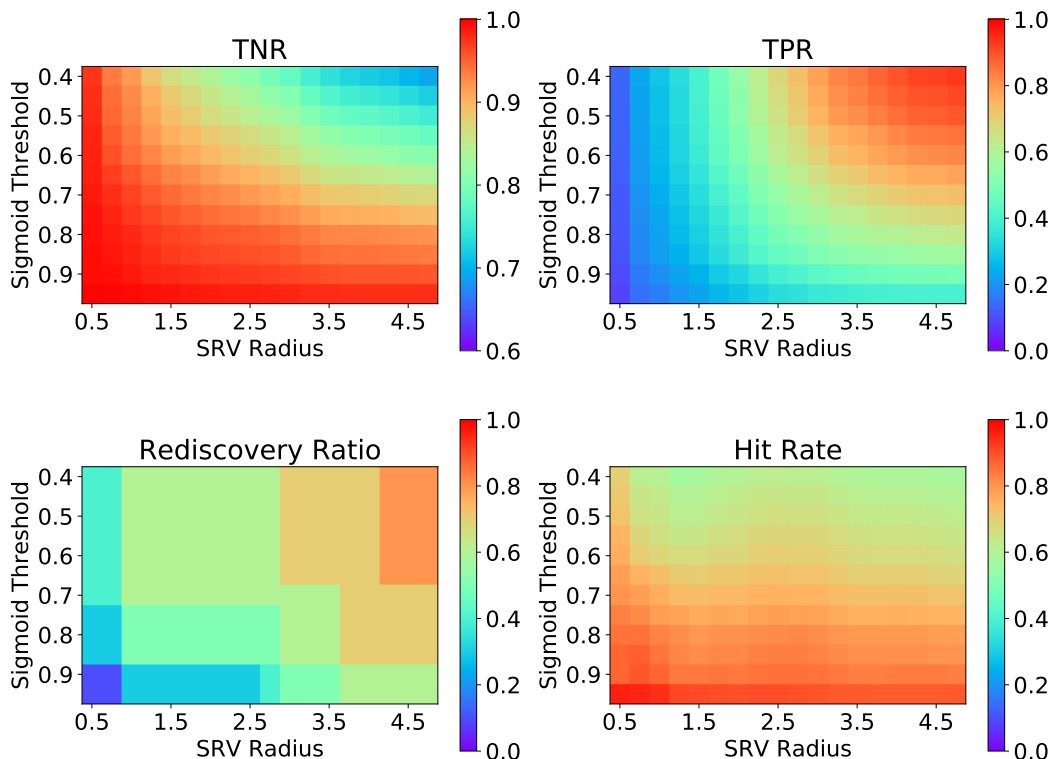

Figure 5: True negative rate (scale starts at 0.6), true positive rate, rediscovery ratio, and hit rate for the extrapolation set 9-12 for varying sigmoid threshold $\tau$ and SRV radius $r$. For too restrictive parameter choices ($\tau \to 1$ and $r \to 0.5$) the TNR approaches 1, while TPR and rediscovery ratio approach 0, such that no interesting new setups would be identified. For too loose choices (small $\tau$, large $r$), too few negative samples would be rejected, such that the advantage over random search becomes negligible. For a large variety of $\tau$ and $r$ the models perform satisfyingly well, allowing a decent compromise between TNR and TPR. This is reflected in large values for the hit rate, which is 0.736 on average over all depicted thresholds.

Finally, we investigate the conventional in-distribution generalization error using the test set (20 % of the data). Entanglement classification: The entanglement training BCE loss value is 10.2. TNR and TPR are $0.9271 \pm 0.00024$ and $0.9469 \pm 0.00041$, respectively. The corresponding test error is 10.4. TNR and TPR are $0.9261 \pm 0.00038$ and $0.9427 \pm 0.00065$, respectively. SRV regression: The SRV training loss value according to Equation (2) is 2.247, the accuracy with $r = 3$ is 93.82 % and the mean distance between label and prediction is 1.3943. The SRV test error is 2.24, the accuracy with $r = 3$ is 0.938 and the mean distance between label and prediction is 1.40. These figures are consistent with a clean training procedure.

## 4 OUTLOOK

Our experiments demonstrate that an LSTM-based neural network can be trained to model certain properties of complex quantum systems. Our approach is not limited to entanglement and Schmidt rank but may be generalized to employ other objective functions such as multiparticle transformations, interference and fidelity qualities, and so on.

Another possible next step to expand our approach towards the goal of automated design of multiparticle high-dimensional quantum experiments is the exploitation of generative models. Here, we consider Generative Adversarial Networks (GANs) (Goodfellow et al., 2014) and beam search (Lowerre, 1976) as possible approaches.

Generating sequences such as text in adversarial settings has been done using 1D CNNs (Gulrajani et al., 2017) and LSTMs (Yu et al., 2016; Fedus et al., 2018). The LSTM-based approaches employ ideas from reinforcement learning to alleviate the problem of propagating gradients through the softmax outputs of the network. Since our data is in structure similar to text, these approaches are directly applicable to our setting.

For beam search, there exist two different ideas, namely a discriminative approach and a generative approach. The discriminative approach incorporates the entire data set (positive and negative samples). The models trained for this work can be used for the discriminative approach in that one constructs new sequences by maximizing the belief of the network that the outcome will be a positive setup. For the generative approach, the idea is to train a model on the positive samples only to learn their distribution via next element prediction. On inference, beam search can be used to approximate the most probable sequence given some initial condition (Bengio et al., 2015). Another option to generate new sequences is to sample from the softmax distribution of the network output at each sequence position as has been used for text generation models (Graves, 2013; Karpathy & Fei-Fei, 2015).

In general, automated design procedures of experiments has much broader applications beyond quantum optical setups and can be of importance for many scientific disciplines other than physics.

## 5 CONCLUSION

We have shown that an LSTM-based neural network can be trained to successfully predict certain characteristics of high-dimensional multiparticle quantum states from the experimental setup without any explicit knowledge of quantum mechanics. The network performs well even on unseen data beyond the training distribution, proving its extrapolation capabilities. This paves the way to automated design of complex quantum experiments using generative machine learning models.

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
