# OpenReview forum: "Quantum Optical Experiments Modeled by Long Short-Term Memory"
_ICLR.cc/2020/Conference — Reject_

### Official Review · AnonReviewer2 · 2019-10-21
**Official Blind Review #2**

**Rating:** 3

**Review:**

This paper proposed to use machine learning models to predict certain properties of complex quantum systems. In quantum physics experiments, one need to randomly search millions of experimental setups to search for interesting experiments. This paper shown that machine learning models can provide significant improvement over random search.

In general, this paper is easy to follow and clearly presents the main idea and verifies the effectiveness of the proposed method. However, I still have some concerns about this paper. This paper only used the machine learning method, i.e. LSTM model and binary cross entropy loss function for solving quantum physical tasks, which provides less knowledge to the machine learning or representation learning community. In my opinion, this paper should be submitted to a quantum physics journal or conference, rather than a machine learning conference.

**Experience Assessment:**

I do not know much about this area.

**Review Assessment: Checking Correctness Of Derivations And Theory:**

I carefully checked the derivations and theory.

**Review Assessment: Checking Correctness Of Experiments:**

I carefully checked the experiments.

**Review Assessment: Thoroughness In Paper Reading:**

I read the paper at least twice and used my best judgement in assessing the paper.

---

### Official Review · AnonReviewer1 · 2019-10-22
**Official Blind Review #1**

**Rating:** 1

**Review:**

This paper investigates how LSTM networks learn quantum optical experimental setup and predict characteristics of resulting quantum states. While the content of this paper may be of interest to quantum computing specialists (I am unable to actually judge that), my impression is that it is out of scope for ICLR. Not by default, but by the style of writing and lack of effort to make the studied problem accessible to non-expert audience. After a general introduction the paper starts immediately with a physics notation for wavefunctions (ok this one could argue is still generic enough for many scientists to understand), but right after is talks of orbital angular momentum and Schmidt rank vector. I do have a background in physics, but not speciality in quantum computing, and at the end of section 2.1 I am already lost. Subsequent section are clearer but since I did not understand how is the data really represented it is impossible for me to gain insight on what is actually being done. The same is true with the evaluation as I do not know what I should be comparing to and what the standard methods give. The paper could have been a nice opening towards new applications for ICLR but it would have to be written in a much more pedestrian manner. In the present form I argue for rejection.

**Experience Assessment:**

I do not know much about this area.

**Review Assessment: Checking Correctness Of Derivations And Theory:**

N/A

**Review Assessment: Checking Correctness Of Experiments:**

N/A

**Review Assessment: Thoroughness In Paper Reading:**

I read the paper at least twice and used my best judgement in assessing the paper.

---

### Official Review · AnonReviewer3 · 2019-10-22
**Official Blind Review #3**

**Rating:** 1

**Review:**

This paper looks at the problem of predicting 2 properties of quantum states produced by an optical table consisting of a sequence of physical elements that modify a quantum state in a particular way, given the sequence of optical elements applied. The authors train 2 separate recurrent networks (LSTMs). The first one takes a sequence of optical elements and predicts a binary boolean label that answers the question of whether the resulting quantum state would be maximally entangled. The second one takes a sequence of optical elements and predicts the Schmid rank vector of the resulting state -- a linear algebraic quantity that is used in quantum information science. The authors explore whether the trained networks generalize to longer sequences of optical elements than trained on, and conduct other checks on the result.

I really appreciate that the authors checked the applicability to sequences of elements longer than trained on. Another great point is the separation of the problem into the two LSTM, so that spurious correlations between the Schmid rank vector and entanglement cannot be used by the NN.

While overall I find the problem interesting and the authors’ approach reasonable, I have a large number of questions / points of confusion that I will detail below.

--------------- Point 1 ----------------
It is unclear what is actually fed into the optical table. My interpretation of the paper is that the input state to the optical table *must be fixed*, since it is not an input to the LSTM. This severely limits the applicability of the framework presented.

I was unable to find what quantum state is actually being input into the sequence of optical elements whose effect you are trying to model. Considering that the properties of the quantum state /after/ the application of the set of N optical elements depend very strongly on the *input quantum state*, I find this omission significant. For an input state rho_0, each element L (which in this particular case is the input to the LSTM), will transform the state as rho -> L rho -> rho’. A series of N elements can be viewed as an application of a sequence of N operations L_1, …. , L_N. Therefore the resulting state is rho_final = L_N L_{N-1} …. L_1 rho_0. As it is clear here, the output state rho_final depends *both* on a) the sequence of applied operations / optical elements (L1,L2, …, LN) as modeled in this paper, and b) the actual input state rho_0. Since what is actually fed into the LSTM is only the sequence of optical elements  (L1,L2, …, LN), while the loss depends on the properties of rho_final, I believe rho_0 must have been assumed fixed and constant throughout the whole paper (i.e. that every data point in the train and test set actually had the same rho_0). If that is the case, the problem that the paper addresses is not /predicting the properties of a quantum state produced by a particular sequence of optical elements/, but rather /predicting the properties of a quantum state produced by a particular sequence of optical elements *given a fixed, constant input state, and not other input state*/. That is a much more restricted regime. I think authors should be clearer about the actual problem their paper is addressing.

--------------- Point 2 ----------------
Lack of baselines. The problem setup and metrics used are quite specific (understandably) to the problem at hand. However, I do not have intuitions for what a “baseline” performance should be and therefore cannot validate whether the approach presented improves in it. I would like to see something equivalent to the 10% chance result on a 10-class classification task. To be more precise, while the binary classification of the maximally entangled nature of the quantum state is clear, the L_2 distance on the sorted 3-tuple of the Schmid rank vector is less so. The explicitly encoded sorted nature of the output 3-tuple (it is parameterized as such) makes it hard for me to make sense of the L_2 distance threshold r that the authors use (which is in fact at some sections r=3, i.e. quite large). I think that adding a baseline performance of some sort would greatly improve my ability to see how much better the LSTM approach performs.

--------------- Point 3 ----------------
The datatype of the LSTM inputs is unclear to me. As I discussed in point 1), the resulting quantum properties of the output state after the application of N optical elements depends on the input state as well as the sequence of elements applied. I am not clear on how the authors encode the optical elements themselves -- i.e. what datatype are the inputs x1, … , xN to the LSTM. If they model them categorically as one-hot vectors, i.e. a 0 degree orientation polarization plate as (1,0,0,0,...0), a beam splitter as (0,1,0,0,....,0), a 90 degrees polarization place as (0,0,1,0,....,0) etc, they are restricting the applicability of the trained models to that particular set of optical elements. For example, if I wanted to know what adding a polarization plate with orientation alpha will do, I would not be able to do that. If it indeed is true that the inputs are drawn from a small number of possible optical elements that has to be specified prior to training, this would again limit the scope of applicability of the results presented.

--------------- Point 4 ----------------
States as state vectors or density matrices? It wasn’t clear to me what the actual state that is being fed into the optical table is. This might be a minor point, but Equation 1 suggests it is in fact a state vector, while I thought quantum optical experiments work with density matrices. This is important for the dimensionality considerations involved, since e.g. for a set of M qubits, the state vector would have 2^M elements (-constraints), while the density matrix would have 2^(2M) elements (-constraints). Solving the latter problem would therefore be more impressive than the former.

--------------- Point 5 ----------------
Are you always working with 3 subsystems/photons? From reading the paper I didn’t understand whether you were restricting your setup to always work in the 3 photon regime that you mentioned in Equation 1. What is the range of the values that can appear in the Schmid rank vector? If they are small, then setting the threshold r=3 would be very generous and making it easy for the LSTM.

--------------- Conclusion ----------------
While I like the problem and the attempted solution, the results presented (as I understood them) are more restricted and therefore weaker than I originally expected. I believe the results are promising, but more clarity and more work on generalizing beyond the restricted regime presented would greatly improve the paper.


**Experience Assessment:**

I have published one or two papers in this area.

**Review Assessment: Checking Correctness Of Derivations And Theory:**

I assessed the sensibility of the derivations and theory.

**Review Assessment: Checking Correctness Of Experiments:**

I assessed the sensibility of the experiments.

**Review Assessment: Thoroughness In Paper Reading:**

I read the paper at least twice and used my best judgement in assessing the paper.

---

### Decision · Program_Chairs · 2019-12-19

**Decision:**

Reject

**Comment:**

The paper predicts properties of quantum states through RNNs.  The idea is nice, but the results are very limited and require more work.  It seems to be more suited for a conference focussing on quantum ML---even when the authors have an ML background.

All reviewers agree on a rejection, and their arguments are solid.  The authors offered no rebuttal.